# Colombian Scorpion *Centruroides margaritatus*: Purification and Characterization of a Gamma Potassium Toxin with Full-Block Activity on the hERG1 Channel

**DOI:** 10.3390/toxins13060407

**Published:** 2021-06-08

**Authors:** José Beltrán-Vidal, Edson Carcamo-Noriega, Nina Pastor, Fernando Zamudio-Zuñiga, Jimmy Alexander Guerrero-Vargas, Santiago Castaño, Lourival Domingos Possani, Rita Restano-Cassulini

**Affiliations:** 1Grupo de Investigaciones Herpetológicas y Toxinológicas, Centro de Investigaciones Biomédicas, Departamento de Biología, Facultad de Ciencias Naturales, Exactas y de la Educación, Universidad del Cauca, Sector Tulcan, Calle 2 N 3N-100, Popayán 190002, Cauca, Colombia; jbeltran@unicauca.edu.co (J.B.-V.); guerrero@unicauca.edu.co (J.A.G.-V.); 2Grupo de Investigación Laboratorio de Herpetología y Toxinología, Departamento de Fisiología, Facultad de Salud, Universidad del Valle, Calle 4B N° 36-00, Santiago de Cali 760043, Valle del Cauca, Colombia; santiago.castano@correounivalle.edu.co; 3Departamento de Medicina Molecular y Bioprocesos, Instituto de Biotecnologia, Universidad Nacional Autónoma de México, Av. Universidad 2001, Cuernavaca 62210, Morelos, Mexico; edson.carcamo@proteininnovation.org (E.C.-N.); fernando.zamudio@ibt.unam.mx (F.Z.-Z.); lourival.possani@ibt.unam.mx (L.D.P.); 4Centro de Investigación en Dinámica Celular, Research Institute in Basic and Applied Sciences (IICBA), Universidad Autónoma del Estado de Morelos, Av. Universidad 1001, Col. Chamilpa, Cuernavaca 62209, Morelos, Mexico; nina@uaem.mx

**Keywords:** *Centruroides margaritatus*, CmERG1, CnERG1, Electrophysiology, ERG toxin, ERG channel

## Abstract

The Colombian scorpion *Centruroides margaritatus* produces a venom considered of low toxicity. Nevertheless, there are known cases of envenomation resulting in cardiovascular disorders, probably due to venom components that target ion channels. Among them, the *human*
*ether-à-go-go-Related* gene (hERG1) potassium channels are critical for cardiac action potential repolarization and alteration in its functionality are associated with cardiac disorders. This work describes the purification and electrophysiological characterization of a *Centruroides margaritatus* venom component acting on hERG1 channels, the CmERG1 toxin. This novel peptide is composed of 42 amino acids with a MW of 4792.88 Da, folded by four disulfide bonds and it is classified as member number 10 of the γ-KTx1 toxin family. CmERG1 inhibits hERG1 currents with an IC_50_ of 3.4 ± 0.2 nM. Despite its 90.5% identity with toxin ɣ-KTx1.1, isolated from *Centruroides noxius*, CmERG1 completely blocks hERG1 current, suggesting a more stable plug of the hERG channel, compared to that formed by other ɣ-KTx.

## 1. Introduction

Scorpions are arachnids of wide geographic distribution, with around 2200 species described in families recognized worldwide [1]. The *Buthidae* family has the most studied venoms due to its great toxicity to mammals and hence to humans [2]. In Colombia, five genera compose this family: *Anantheris* (13 species), *Centruroides* (4 species), *Microtityus* (2 species), *Rhopalurus* (1 species), and *Tityus* (13 species). The four species of the genus *Centruroides* in Colombia are: *C. eduardsii*, *C. gracilis*, *C. marx*, and *C. margaritatus* [3]. *C. margaritatus* is distributed in two geographically isolated areas: the upper and middle basin of the Cauca River (Valle del Cauca, Colombia) and the Patía river Valley (Cauca, Colombia) [4,5].

Scorpion venom is a mixture of proteins, peptides and enzymes, carbohydrates, free amines, nucleotides, lipids, and other low molecular weight components with unknown function. Peptides that act as ion channel modulators are the main agents responsible for the venom toxicity and they have been classified according to their targets into: sodium scorpion toxins (NaScTx), with molecular masses between 6–8 kDa [6], potassium scorpion toxins (KScTx), with molecular masses between 3–5 kDa [7,8], and calcium scorpion toxins (CaScTx) that comprise peptides acting on voltage gated calcium channels and that specifically modulate ryanodine receptors [9,10]. In the last decades, many details of the toxin-channel interaction have been clarified and models of different mechanisms of toxin binding have been described [11,12,13]. Based on their structural and functional characteristics, KScTx have been classified into seven subfamilies: α-Ktx, β-Ktx, ɣ-KTx, δ-KTx, Ɛ-Ktx, κ-KTx, and ʎ-KTx (kalium database) [14]. The ɣ-KTx family comprises toxins that selectively bind to ERG (*Ether-à-go-go-Related* Gene) potassium channels. These channels are expressed in many tissues and they are especially important for the repolarization of the cardiac action potential. Mutations in the erg1 gene are responsible for congenital long QT syndrome, a disorder of cardiac repolarization, which is characterized by prolongation of the QT interval on the surface electrocardiogram, abnormal T waves, and risk of sudden cardiac death due to ventricular arrhythmias [15]. The first member of the ɣ-KTxs was isolated from the venom of the scorpion *Centruroides noxius* and named CnERG1 (ErgTx1, ɣ-KTx1.1) [16]. Thereafter, many ɣ-KTxs sequences were identified from scorpions of the genus *Centruroides* and *Mesobuthus* [17,18,19], and the toxin-channel interactions were characterized for some of these peptides [20,21,22]. A common feature of these toxins is that, despite their concentration, ERG channel blocking effect is always partial (about 90% for CnERG1) [23] and that ɣ-KTxs accelerate the closure kinetics due to their preference for the channel closed state [22]. These facts have been partially explained by proposing that the ɣ-KTxs-ERG channel interaction is of the “turret” type, where toxins interact with the zone of the extracellular loop between the transmembrane segments S5 and S6, also called the “turret” of the channel [24,25].

*C. margaritatus* is a markedly synanthropic species [5,26] that produces a venom of low toxicity with LD50 of 59.9 mg/kg [5]; however, in scorpion stings by *C. margaritatus*, there have been reports of clinical symptoms associated with cardiovascular disorders, leading even to scorpionism with moderate and severe systemic manifestations [4,27]. Previous studies using rats as biological models showed that intravenous administration of a chromatographic fraction (peptides between 2.5 and 6.0 kDa) of the *C. margaritatus* venom caused important cardiovascular alterations that included hemodynamic failure. In addition, the histological analysis showed a high density of interfibrillar hemorrhage in cardiomyocytes exposed to the venom fraction [28,29]. All these alterations induced by the *C. margaritatus* venom may be associated with toxins that interact directly in the heart or smooth muscle, with sodium or potassium channels and specifically with the ERG potassium channels.

Until now, little is known about the *C. margaritatus* venom composition. Margatoxin 1, an α-KTx of 39 amino acids and three disulfide bonds, was earlier identified as a potent inhibitor of the Kv1.3 channel in human peripheral T lymphocytes [30] and later found to inhibit also Kv1.1 and Kv1.2 channels with similar affinity [31]. Thereafter, in a proteomics study of *C. margaritatus* venom, two other peptides were isolated and characterized: a peptide with 24 amino acids and 3 disulfide bridges (MW = 2609.15 Da) and a peptide with 30 amino acids and 3 disulfide bridges (MW = 3376 Da), both classified as αKTxs [32]; however, no function was tested for these peptides.

In this work, we describe the characterization of *C. margaritatus* venom in order to determine its activity on voltage-gated sodium and potassium channels. In addition, we present venom separation aimed to identify potassium channel toxins able to block the hERG1 potassium channel. A new γKTx toxin (CmERG1 or γKTx1.10) is described and characterized. CmERG1 was sequenced and the differences in structure and functional features with CnERG1, as well as toxin-channel interaction models, are discussed.

## 2. Results

### 2.1. Peptide Isolation

The *C. margaritatus* soluble venom was first separated by using reversed-phase (RP) chromatography using a fast protein liquid chromatography (FPLC) system, resulting in 41 fractions (Figure 1A). The molecular masses of the principal fractions were evaluated by electrospray ionization mass spectroscopy (ESI-MS) obtaining a partial fingerprint of the main components (Table 1). Peptides with molecular masses between 3–5 kDa correspond to peptides that are mainly active on potassium channels; therefore, the fractions with molecular masses in this range were selected to be evaluated by electrophysiology on hERG1 channels. From this first screening, we found that fraction 23 (containing mainly the peptides of molecular masses 6838.9 and 4792.1 (Table 1)) fully inhibits the hERG1 currents.

A further purification of F23, by reversed-phase high performance liquid chromatography (RP-HPLC), resulted in the isolation of a peptide corresponding to 4792.88 Da (Figure 1B). Subsequent electrophysiology analysis confirmed the activity of this peptide against hERG1. Due to the scarce amount of the peptide of MW 4792, we performed a preparative purification using a three-step strategy aimed at obtaining a greater amount of the isolated peptides, enough to perform the sequence determination and the electrophysiological characterization. The first step was gel-filtration chromatography in a Sephadex G-50 column, resulting in three fractions (FI, FII and FIII, Figure 1C). According to previous separations of other scorpion venoms [32,33], we know that fraction FII usually contains the toxic components of the venom, so we directly separated fraction FII through cation-exchange chromatography, using a carboxymethyl cellulose (CMC) column. The resulting 10 sub-fractions (Figure 1D) were individually separated by a C18 column, and the molecular masses were determined for the principal peaks by using ESI-MS (Table 2). The peptide with MW 4792.88 Da was found in FII.6 (Figure 1D,E), and it corresponds to a new toxin called CmERG1. The complete sequence of the CmERG1 peptide (Figure 2) was obtained by automatic Edman degradation, using the conditions described in Material and Methods.

### 2.2. CmERG1 Primary Structure

Primary structure determination of the CmERG1 was achieved by direct Edman degradation of the native peptide and of the reduced and alkylated peptide. CmERG1 is a toxin consisting of 42 amino acids with eight cysteines and four putative disulfide bonds:

DRDSCVDKSRCAKYGYFQECTDCCKKYGHNGGTCMFFKCKCA.

We analyzed the amino acid sequence of CmERG1 with the Basic Local Alignment Search Tool (BLAST) of the National Center for Biotechnology Information (NCBI): CmERG1 belongs to the γKTx1 potassium toxin family and is member γKTx1.10 of this toxin family (it can be found in the UniProt Knowledgebase with the accession number C0HLM3). CmERG1 has 90.5% identity with CnERG1 (γKTx1.1), the first ERG toxin isolated from *C. noxius*, and it shows 85–97% identity with the other γKTx1 members (Figure 2). It differs in three highly conserved residues amongst these toxins and it is interesting to notice that the modification Y17F is in a position referred to be important for the CnERG1-hERG1 interaction (Figure 2, residues highlighted in magenta).

### 2.3. CmERG1 Block Action on hERG1 Channels

The effect of the CmERG1 toxin was evaluated on the hERG1 channel by means of electrophysiology experiments. Similar to the action of CnERG1 on hERG1, CmERG1 blocks channels in a fast and reversible manner (Figure 3C). Dose response curves were calculated by application of toxin concentrations from 1 nM to 1 µM. Data were fitted with a logistic equation resulting in an IC_50_ of 3.4 ± 0.2 nM and slope of 1.1 ± 0.05 (Figure 3A). Surprisingly, at the highest concentrations (1 µM), CmERG1 was able to almost completely block the channels. This feature is different from the other γKTxs, like CnERG1, that allow passage of about 10% current even at saturating concentrations. Moreover, the CmERG1 toxin does not accelerate the kinetics of the closing process (Figure 3B). These differences between the CmERG1 toxin mode of action and that of other γKTxs suggest that this toxin may interact with ERG1 channels forming a more stable pore block than other γKTxs, which could engage more frequently with the channel turrets [23].

### 2.4. CmERG1 Does Not Modify the Activation and Inactivation Kinetics of hERG1 Channels

To understand better the activity of CmERG1 on hERG1, we analyzed the voltage dependence of the activation and inactivation processes in presence or absence of CmERG1 at 4 nM, a concentration close to the IC50. For the activation, tail currents were elicited at −120 mV after 5 s preconditioning pulses from 30 to −80 mV (Figure 4A). Figure 4B,C illustrates typical currents recorded in the absence and in the presence of CmERG1. Tail currents were normalized, plotted versus preconditioning potentials, and fitted to a Boltzmann equation (Figure 4D). The toxin reduces the total current with a tendency to modify the potential of half activation, that was −37.5 ± 1.4 mV in control and -34.8 ± 0.8 mV in presence of the toxin. A slight change was also observed for the slope that was 28.2 ± 2.2 in the absence and 29 ± 1 in the presence of toxin (*n* = 4, paired sample *t*-test at 0.05 level).

For steady state inactivation, tail currents were recorded during 500 ms depolarization steps from −170 to 40 mV, after a depolarization at 40 mV for 1 s (Figure 4E). Figure 4F,G show an example of currents recorded in the absence and in the presence of toxin. Peak currents were corrected for deactivation as described in the inset. Corrected currents were converted to conductance, normalized, and plotted versus voltage (Figure 4H). Black lines are the best fit obtained by a Boltzmann equation giving V_1/2_ for inactivation of −77.6 ± 3.6 mV and −79 ± 1.3 mV, and slope of 28.2 ± 2.2 and 29 ± 1.5 for control and in the presence of 4 nM CmERG1, respectively. The toxin does not induce significant changes on the voltage dependence of inactivation (*n* = 5, paired sample *t*-test at 0.05 level).

### 2.5. Models of Channel-Toxin Interactions

As shown in Figure 2, CmERG1 and CnERG1 are 90% identical in sequence, and, except for residue 17, share the same amino acids at the positions involved in hERG1 binding [20,21]. The most critical residues, resulting in >100-fold decrease in toxin potency when mutated to alanine are K13, Y14, F37, and Y17; M35 and Q18 have more modest effects. We also surmised that any difference in action shown by these two toxins should be a consequence of the four differences located at positions 17, 21, 26, and 27. Assuming that these four differences are a perturbation to the main binding mode, based on the fact that they lie along the alpha-helix of the toxin, away from the critical residues for binding, we considered as a first approximation that both CmERG1 and CnERG1 use the same interaction surface when binding to the channel; this assumption will be tested by mutagenesis studies of CmERG1 in the future. Therefore, we carried out docking assays with HADDOCK 2.4 [33,34], imposing as flexible constraints that residues 13, 14, 17, 18, 21, 26, 27, 35, and 37 in the toxins interact with residues 582, 583, 585, 588, 592, and 628 in the channel [24,35]. Given that these constraints are flexible, we found many toxin-channel complexes, for both toxins, with incomplete blocks of the channel, probably corresponding to the “turret block” discussed in the literature. Intriguingly, we also find two blocking modes, shown in Figure 5A,B. The configuration with the best interaction energy corresponds to Figure 5A, where a “cationic-hydrophobic plug” interacts directly with G628 at the entrance of the selectivity filter; this plug was proposed [36] when the NMR structure of CnERG1 was determined, and this binding mode is consistent with the fact that it is fairly insensitive to ionic strength, suggesting that the hydrophobic interactions are important. This plug is comprised of K13, flanked at three sides by Y14, F36 and F37. A less stable complex is depicted in Figure 5B, where the toxin is off-center with respect to the selectivity filter, which is now blocked by Y14. Both types of block were found for CmERG1 and CnERG1. Figure 5C shows a top view of a high affinity CnERG1-hERG complex, with an almost perfect alignment of the toxin helix and the center of the selectivity filter; note that accommodating a dimer of toxins would not the feasible in this configuration, as they would not fit.

Taking the channel-toxin complexes with a blocking configuration and the best binding energy, we compared the local environment (within 4.5 Å) of the residues critical for binding of CnERG1 and the four positions where CmERG1 and CnERG1 differ. These are shown in Figure 6 (for CmERG1) and Figure 7 (for CnERG1). We discuss similarities and differences in two groups: those residues that block the selectivity filter and those that contribute to the “seal” between the toxin and the channel, located in the turrets. Positions 26 and 27 do not engage in interactions with the channel residues.

In the first group we find K13, Y14, and F37. K13 engages in the same interactions for both toxins (Figure 6A and Figure 7A). Y14 interacts with the floor of the pore in both cases but differs mainly in its interaction with Q592 in CmERG1 (Figure 6B) and with W585 in CnERG1 (Figure 7B). F37 interacts with W585 in both toxins, but lies closer to the floor of the pore, interacting also with G628 in CmERG1 (Figure 6C); in CnERG1 it interacts more with turret residues N588 and Y597 (Figure 7C). This could be one of the reasons for the lability of CnERG1 in blocking the current completely.

The “seal” residues are F/Y17, Q18, T/Q21, and M35. F17 (Figure 6D) and Y17 (Figure 7D) engage in the same type of interaction with Y597, suggesting that the effect of losing/gaining an -OH moiety is minimal. Q18 lies at the frontier of two channel subunits. In CmERG1 it stacks with its amide group against the peptide bond between residues P596 and Y597, while engaging in a hydrogen bond with R582 from the neighboring subunit (Figure 6F); in CnERG1 it also stacks against the channel, but now against Y597 and the backbone of K595, while hydrogen bonding to H578 and R582 from the adjacent subunit (Figure 7E). T21 in CmERG1 does not interact with the channel, while Q21 appears to engage in aromatic interactions of its sidechain amide with its neighbors in the channel (Figure 7F). Finally, M35 engages in more interactions with turret residues in CnERG1 (Figure 7G) than in CmERG1 (Figure 6E). This increase in interactions with turret residues in CnERG1 compared to CmERG1, combined with the weaker interaction of F37 with the entrance of the selectivity filter, could explain the 10% difference in efficiency at a total block of current between both toxins.

## 3. Discussion

The venom of Buthidae scorpions is rich in neurotoxins acting on voltage-dependent ion channels that are responsible for symptomatologic signs in autonomic hyperactivity (tachycardia, hypertension, cardiac arrhythmias, mydriasis, excessive salivation and tearing, bradycardia, hypotension, and others) [4,37,38]. When these accidents occur, they might cause death, as a consequence of cardiovascular defects attributed to the massive release of catecholamines from the adrenal glands and noradrenergic nerve terminals, together with complications associated with pulmonary edema and respiratory failure in mammals [4,39,40].

Despite the fact that the *C. margaritatus* scorpion belongs to the *Buthidae* genus, its venom has been considered of low toxicity [1], but as mentioned above, in Colombia this species has caused a few cases of moderate and severe scorpionism [4]. In Colombia, two populations of this species are differentiated according to the geographical distribution: the first in the Patía Valley with a LD50 of 42.83 mg/Kg calculated in mice by intraperitoneal administration [29] and that is the object of this study; the second, in the Cauca Valley with an LD50 of 59.9 mg/kg calculated in mice by intraperitoneal administration [5]. Until now, scarce information was available for *C. margaritatus* venom; the only venom component characterized by electrophysiology assays was Margatoxin (α-KTx2.2), isolated from *C. margaritatus* venom and first identified as a potent and specific blocker of Kv1.3, but later discovered to block also Kv1.1 and Kv 1.2 [30,31]. Moreover, studies in rats showed that administration of fractions of the *C. margaritatus* venom caused cardiovascular alterations [28,29,41], but which particular component is responsible for these effects is unknown.

Here we analyzed the composition of *C. margaritatus* venom looking for peptides active on hERG1 potassium channels, that could explain the cardiovascular alterations reported as consequences of *C. margaritatus* stings and also reproduced in rats as biological models [28,41]. Potassium hERG1 channels are involved in cardiac physiology, where they contribute to the repolarization of cardiac action potentials; therefore, alteration in ERG currents is associated to arrhythmias and cardiac failure [42]. We found in the *C. margaritatus* venom a new peptide acting on the hERG1 channel. It is a 42 amino acid peptide with four disulfide bonds, which was given the trivial name of CmERG1. According to the international classification [43], CmERG1 corresponds to the systematic code of ɣ-KTx 1.10, and it appears in the UniProt Knowledgebase under the accession number C0HLM3.

Unlike the other ɣKTxs, the new toxin CmERG1 has the ability to block 100% of the hERG1 current. The nature of the CmERG1 full blockage activity is unknown but certainly resides in its sequence. The mechanism of action of γKTxs has been proposed to be through interactions with the channel “turret” [22,24,35], where the S5-pore loop conforms an aliphatic α-helix [44] that makes contacts with a cationic and hydrophobic patch in the γ-toxins surface. In CnERG1, the hydrophobic patch is situated at the N-terminal end of the α-helix and the β sheet side of the toxin [36,45]; furthermore, site-directed mutagenesis to alanine has confirmed that amino acids K13, Y14, Y17, Q18, M35, and F37 are important for the CnERG1-hERG1 interaction [20,21].

Two models were proposed to explain the uncommon feature of the γKTxs that comprises a partial current block, despite the saturation concentration, and the acceleration of the closing process due to the toxin preference for the channel closed state. In one kinetic model proposed by Hill and collaborators [23], first the toxin binds to the channel in a toxin channel encounter complex permissive to ion flow (TC*) and then passes to the blocked toxin-channel complex (TC). The incomplete blockage is explained by the relatively rapid rate of dissociation of the complex (TC) compared to the conversion rate of the toxin channel meeting complex (TC*) to TC [16,24,35]. In addition, Tseng’s group proposed for BeKm-1, a “state-dependent” channel-toxin interaction, in which toxin binds preferentially to the closed channels while inactivation promotes toxin unbinding [22,46]. This explains the acceleration of deactivation process first observed by Tseng’s and collaborators and the γKTxs proclivity to unbinding at depolarized potential. Actually, these two models are not mutually exclusive and may coexist as we have previously seen with CeErg4 [47].

CmErg1 eliminates the current by 100%, suggesting a higher affinity for the pore-blocking configuration over the turret-binding one. We propose qualitative binding models of the pore-blocking configurations of CmERG1 and CnERG1 for comparison. Following Hill and collaborators, in the mindset of an equilibrium between the turret-binding mode and the pore-binding mode, the detailed analysis of the interactions of the toxins with the channels point to small differences in the interactions of F37 (a conserved residue, adjacent to the entrance of the selectivity filter), Q21 (a residue that is a T in CmERG1), and possibly Y17 (a residue that is a F in CmERG1) as the leading candidates to explain the higher efficacy of CmERG1 in blocking the channel: CnERG1 engages in more interactions with the turrets, facing the entry to the selectivity filter at an angle, weakening the interaction with the selectivity filter. The better interactions overall of CmERG1 with the pore, compared to those of CnERG1, bias the equilibrium to the blocking configuration. This leads to a concrete hypothesis testable by mutagenesis: mutating Q21 to T and Y17 to F in CnERG1 should make it a better blocker, and mutating T21 to Q and F17 to Y in CmERG1 should impair its blocking capability.

## 4. Conclusions

In this study, we reported the isolation, determination of the primary structure, and electrophysiological evaluation of the new CmERG1 peptide. This work has the weakness of a low number of cells we have been working with in electrophysiology (in some cases *n* = 3 or 4), nevertheless our results suggest that the CmERG1 toxin-channel interaction resembles the blocking characteristics of the α-KTxs with a full blocking, absence of deactivation process acceleration and no modification of the steady state inactivation. In addition, we have proposed specific mutations that should eliminate the complete blocking of hERG1 by CmERG1, and reciprocal mutations that could generate a complete block by CnERG1.

## 5. Materials and Methods

### 5.1. Venom Source, Chemical and Reagents

Fifty scorpions of the species *Centruroides margaritatus* were collected in the Patía Valley, municipality of Balboa, Department of Cauca, south west Colombia (2°01′21″ N, 77°10′43″ O, 800 masl), with official permission from National Environmental Licensing Authority (ANLA), Colombia (R. 0152 from 15 February 2015). Venom was obtained by electrical stimulation of the telson, dissolved in water and then centrifuged at 15,000 RPM and 4 °C for 15 min. The supernatant was lyophilized and kept at −20 °C until use. Venom concentration was estimated by absorbance measured at 280 nm with spectrophotometer Nanodrop (TermoFisher Scientific, Waltham, MA, USA). All chemicals and reagents were analytical grade substances and Mili Q water was used through all the procedures.

### 5.2. Peptide Purification

The venom was fractionated by means of FPLC high performance liquid chromatography model NGC chromatography System, Chrom lab Model software, and Bio Frac automatic fraction collector (Bio Rad, Hercules, CA, USA). RP-FPLC was coupled to a reversed phase C18 column, at wavelength λ = 280 nm, through a 60% mobile phase gradient: Solution A (water + 0.10% TFA) and Solution B (acetonitrile + TFA, 0, 10%), with flow rate of 1 mL/min, for 60 min. Peaks were manually collected obtaining 41 fractions (Figure 1A). Each peak or fraction was subjected to protein quantification by Nano Drop Spectrophotometer ND 1000 equipment and then lyophilized using a Savant SC210A Speed vac Concentrator dryer, (Savant Instruments, Inc. now Thermo Fisher Scientific, Waltham, MA, USA). Peaks with activity against potassium channels were further separated by reversed phase at a high-performance liquid chromatography equipment (RP-HPLC), using an analytical C18 reverse-phase column (Vydac, Hysperia, CA, USA). The pure peptides were obtained by a linear gradient from 100% of solution A (0.12 (*v*/*v*) trifluoroacetic acid (TFA) in water) to 60% of solution B (0.10 (*v*/*v*) TFA in acetonitrile) in 60 min at 1 mL/min flow rate. The detection was monitored by absorbance at λ = 230 nm and components were manually collected, dried using a Savant SpeedVac dryer and storage at −20 °C until used for chemical and functional characterization.

From the first separation, we obtained a small amount of toxin CmERG1 that was not enough (in amount and purity) for sequence determination and the electrophysiological characterization. To obtain more quantity of the toxin, we performed a second venom purification with a three-step strategy. This three-step purification allowed us to process more amount of venom and achieve greater yield and purity by using not only reversed-phase chromatography as purification principle but also size exclusion and ion-exchange chromatography. For this, we first applied the soluble venom for the gel filtration on a Sephadex G-50 column (60 cm × 26 mm, L × I.D), in 20 mM ammonium acetate buffer, pH 4.7 at a flow 2 mL/min, and we obtained three fractions. From our experience, the main toxic components are in fraction FII, so it was applied to ion-exchange purification on a carboxy-methyl-cellulose (CMC) column (5 cm × 15 mm, L × I.D.) equilibrated with the same buffer. Chromatography was conducted at a flow rate of 2 mL/min with a linear gradient (0–100%; in 200 min) of 500 mM ammonium acetate buffer pH 7.4. Ten fractions were obtained and dried by lyophilization. All fractions from CMC chromatography were further separated by RP-HPLC, under the same conditions described above. We used an analytical C18 reverse-phase column (Vydac, Hysperia, CA, USA). The pure peptides were obtained by a linear gradient from 100% of solution A (0.12 (*v*/*v*) trifluoroacetic acid (TFA) in water) to 60% of solution B (0.10 (*v*/*v*) TFA in acetonitrile) in 60 min at 1 mL/min flow rate. The detection was monitored by absorbance at λ = 230 nm and components were manually collected, then dried using a Savant SpeedVac (Savant Instruments, Inc. now Thermo Fisher Scientific, Waltham, MA, USA) dryer and storage at −20 °C until used for chemical and functional characterization.

### 5.3. Mass Spectrometry and Sequence by Edman Degradation

Single peaks from RP-HPLC were analyzed by mass-spectrometry (MS) with an electrospray ionization (ESI) equipment LCQ FLEET from Thermo Fisher Scientific Inc. (San Jose, CA, USA). Automatic amino acid sequencing of CmERG1 was performed by Edman degradation using a Biotech PPSQ-31A Protein Sequencer equipment from Shimadzu Scientific Instruments, Inc. (Columbia, MD, USA). A sample of native peptide was directly loaded for sequencing. Additionally, a reduced and alkylated sample of the same peptide was sequenced for identification of the cysteine residues.

### 5.4. Reduction and Alkylation

For reduction, the pure peptide was dissolved in 200 mM TRIS-HCl buffer, pH 8.6 containing 1 mg/mL EDTA and 6 M guanidinium chloride with 2 mg of dithiotreitol (DDT). Nitrogen was bubbled to the solution for 5 min and incubated 45 min at 55 °C. Immediately after 2.5 mg of iodoacetamide was added to the reacting vial, placed in the dark at room temperature for 30 min. Reduced an alkylated peptide was recovered by RP-HPLC (similar conditions as described above).

### 5.5. Amino Acid Sequence Comparison of Peptide

The CmERG1 sequence data reported in this paper appears in the UniProt Knowledgebase under the accession number C0HLM3.

Basic Local Alignment Search Tool (BLAST) by National Center for Biotechnology Information (NCBI) was used to generate the protein sequence alignment in Figure 2.

### 5.6. Electrophysiology

#### 5.6.1. Cells and Solutions

CHO cells stably expressing hERG1 potassium channels were used for the electrophysiological experiments (hERG1 accession number: NP_000229). We previously prepared this cell line stably transfecting CHO cells with plasmid pcDNA3.1-hERG1 (a kind gift from Enzo Wanke from University of Milano-Bicocca, Italy). Briefly: CHO cells at 80% confluence in a 35 mm culture plate were transfected with 2 µg of pcDNA3.1-hERG1 mixed with 7 uL of Lipofectamine (Invitrogen), accordingly to the manufacturer instructions. After 3 days, cells were selected by adding to the culture medium 2 mg/mL of G418 (SIGMA). After 10 days of selection, cells were cloned by limiting dilution and the resulting clones were probed for their current expression by electrophysiological recordings. High glucose DMEM (Dulbecco’s Modified Eagle Medium, SIGMA, Naucalpan de Juarez, Edo de Mexico, Mexico), supplemented with 10% Fetal Bovine Serum (Biotecfron, Emiliano Zapata, Morelos, Mexico) and with 500 µg/mL of antibiotic G418 (SIGMA) was used as growth medium. Cells were routinely maintained at 37 °C with 5% of CO_2_ in humidified atmosphere.

Intracellular solution contained in mM: 130 K-Aspartate, 10 NaCl, 2 MgCl_2_, 10 HEPES, 10 EGTA, pH 7.3 adjusted with NaOH. Extracellular solution contained in mM: 95 NaCl, 40 KCl, 2 CaCl_2_, 2 MgCl_2_, 10 HEPES, 5 glucose, pH 7.3 adjusted with NaOH. A concentrated 100–1000× stock was prepared dissolving lyophilized toxin in distilled water and stored at −20 °C until the used (no more than 3 weeks). Solutions were delivered to the cell under patch by means of an active perfusion system connected to a variable speed syringe pump (model A-99 from Razel Scientific Instruments (Saint Albans VT, USA)). The perfusion rate was approximately 1µL/s.

#### 5.6.2. Patch-Clamp Recordings and Data Analysis

Patch pipettes were manufactured from capillary borosilicate glass tubing (Warner Instruments, Hamden, CT, USA) by means of a vertical puller model P-30 (Sutter Instrument, Novato, CA, USA). When pipettes were filled with internal solution, pipette resistance was between 1.5–3 MOhm.

During the depolarizing steps, hERG1 currents are usually small (Appendix A). This is a consequence of the slow activation kinetics (in seconds) and the very fast inactivation kinetics (in milliseconds) of this channel that can be considered an inward rectifier [48]. Therefore, it is a common practice to record the tail currents during the hyperpolarization, where the inactivation is quickly removed and the deactivation occurs slowly. For the dose response curve (Figure 3), tail currents of hERG1 were recorded at −120 mV for 500 ms, after a preconditioning pulse at 60 mV for 500 ms every 5 s. Peak currents recorded in absence and in presence of the toxin were normalized to the maximal peak current in control condition and plotted versus the logarithm of toxin concentration. Data were fitted by a logistic equation. For the voltage dependence of activation study, hERG1 tail currents were recorded during a step at −120 mV for 500 ms, preceded by a 5 s depolarization steps in the range between 30 to −80 mV. To determine the voltage dependence of inactivation, currents were recorded after a depolarization at 40 mV for 1 s, during depolarization steps in the range between 40 to −170 mV. Currents recorded at potentials lower than −90 mV were corrected for deactivation: the deactivation process was fitted by a single exponential extrapolated to the zero point of each step. Data from both activation and inactivation protocols, in absence and in presence of the toxin, were normalized, plotted versus membrane potential, and fitted by a Boltzmann function.

Currents were acquired by using MultiClamp 700A amplifier and DigiData 1440a (Molecular Devices, San Jose, CA, USA). Off-line analysis and graphs were performed by using Clampfit 10 (Molecular Devices) and Origin 8 (OriginLab, Northampton MA, USA).

#### 5.6.3. Statistical Analysis

Where it is not otherwise indicated, electrophysiological data represent the mean of 3–6 cells ± standard error (S.E.). Each cell was recorded in absence and in presence of the toxin and the difference between these two conditions was analyzed by means of the paired sample *t*-test at 0.05 level.

### 5.7. Modeling of the Toxin-Channel Encounter Complexes

The CmERG1 structure was modeled by homology using the CnERG1 structure available in the Protein Data Bank [49] (PDB-ID 1NE5, chain A) as template in Swiss-Model [50], to ensure correct formation of the four characteristic disulfide bonds of the toxin (between residues 5–23, 11–34, 20–39, and 24–41). The sequence identity between the two toxins is 90%, resulting in a good quality model with GMQE = 0.99 and QMEAN = −1.1. Both CmERG1 and CnERG1 structures were submitted to Charmm-GUI [51] to add hydrogen atoms and generate CHARMM45 [52,53] inputs for energy minimization, which consisted in 200 steepest descent steps for hydrogen atoms only, followed by 400 steepest descent and 400 adopted-basis Newton–Raphson steps for all the atoms.

The structure of the channel, solved by cryoelectron microscopy [54] for the open state, displays the classical tetrameric arrangement of subunits for potassium channels, but without domain swapping; it also lacks coordinates for some residues in the turrets and the extracellular loops. A recent model for the channel, including critical residues in the P-loop (N598-L602 and H578-R582) but lacking still the extracellular loops in the voltage sensor, was proposed to study both blockers and activators of the channel that bind to the intracellular vestibule [55]. In addition, work by Noskov’s group has generated refined models for hERG1 starting from the cryo-EM structure of the channel [54], looking at specific details of the voltage sensors, inactivation mechanisms, and binding sites for drugs [56,57,58]. These are open channels, and no specific attention has been paid to the conformation of the turrets. An independent channel model was kindly donated by Dr. Tseng [22]; this model was based on voltage-dependent potassium channel KvAP and has domain swapping. The structure shows perfect rotational symmetry, so it corresponds to the channel before interacting with BeKm1. Similar, domain swapped models, have been used recently to study the interaction with other toxins [18]. In order to avoid the issue of the position of the extracellular loops of the channel, for which there is no experimental data available, we carried out our docking assays over residues 545 to 668, covering S5, the P-loop and, the S6 helix. The Tseng model was used without further modification. This model superimposes correctly onto structure 5VA1 [54], for the available residues in the turret region.

Docking was carried out in the HADDOCK2.4 server [33,34], using as active residues 582, 583, 585, 588, 592, and 628 in the channel model, and residues 13, 14, 17, 18, 21, 26, 27, 35, and 37 in the toxins. The best structures of each cluster (as indicated by HADDOCK) were inspected in the CASTp3.0 server [59] to select those that interrupted passage through the selectivity filter, using probes of 0.9 and 1 Å radii. These structures were then inspected in VMD [60] to determine the mode of interaction, and to find those that provided the best plugs for the channel pore. These structures were prepared in Charmm-GUI and energy-minimized in CHARMM45 with 300 steepest descent steps for hydrogen atoms only, 500 steepest descent steps for all atoms, and 500 adopted-basis Newton-Raphson steps for all the atoms. Over these structures, the contacts between toxin and channel were calculated with a 4.5 Å cutoff and the hydrogen bonds were calculated with default values (2.4 Å distance, no angle restriction). The list of contacts is rendered in Figure 6 and Figure 7.

## Figures and Tables

**Figure 1 toxins-13-00407-f001:**
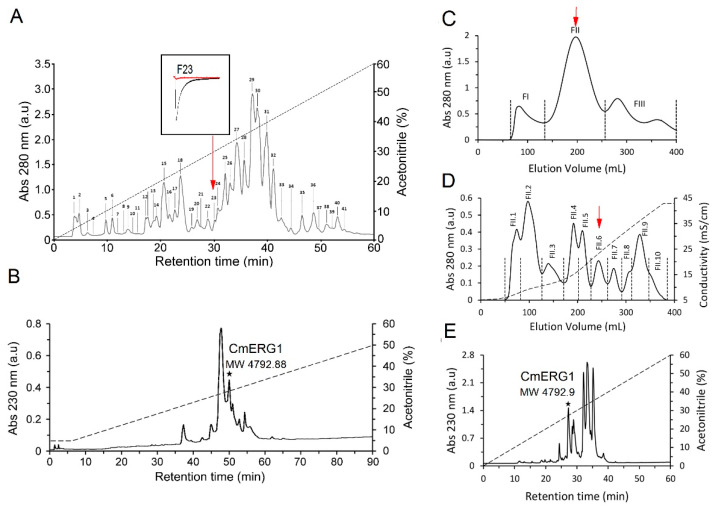
Purification of potassium channel toxins from the venom of *Centruroides margaritatus*. (**A**) Venom separation by RP-FPLC resulting in 41 fractions. Fraction F23 is active on hERG1 channels and is indicated by the red arrow. The inset shows an example of the hERG1 current (black line) and the blocking effect of F23 (red line). (**B**) Re-purification of the fraction F23; the pure peptide active on hERG1 is indicated by the asterisk and corresponds to the one with MW 4792.88 Da and called CmERG1. (**C**–**E**) Isolation of CmERG1 by means of a three-step protocol. First, gel filtration leads to three principal fractions FI, FII and FIII (**C**). From these, fraction FII was further separated by cation-exchange chromatography into 10 sub-fractions (**D**). Toxin CmERG1 was isolated by RP-HPLC from fractions FII.6 with a retention time of 27.4 min (indicated with a star in E). In C and D, the red arrow indicates the fraction separated in the subsequent step.

**Figure 2 toxins-13-00407-f002:**
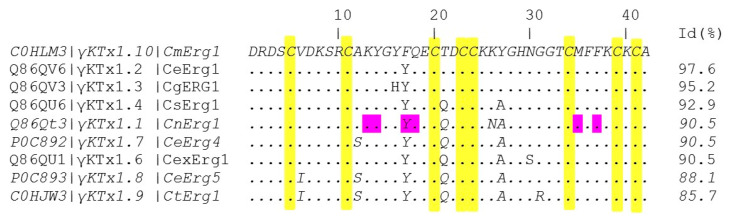
Alignment of CmERG1 with similar ERG toxins from the family γKTx1. Cysteines are highlighted in yellow. The important residues for the CnERG1-hERG1 channel interaction, according to [19,20], are indicated in magenta. Points indicate identical positions to those in CmERG1.

**Figure 3 toxins-13-00407-f003:**
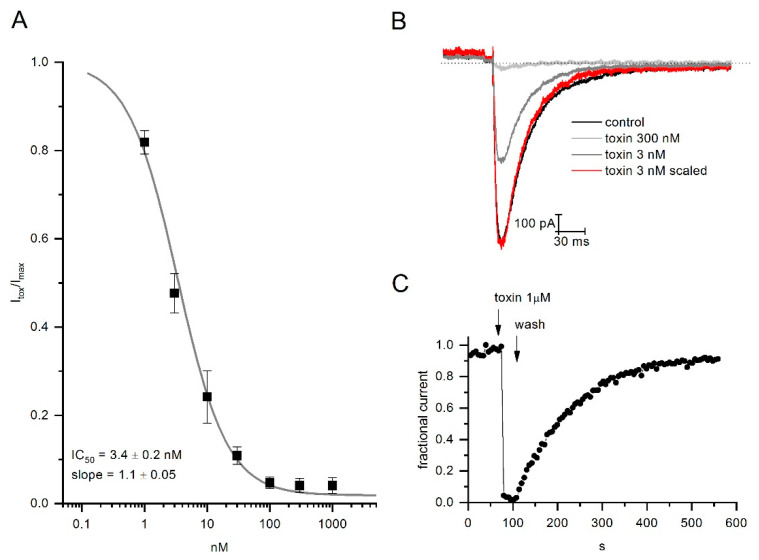
CmERG1 toxin blocking features. (**A**) Dose-response curve. Normalized peak currents were plotted versus the logarithm of toxin concentration. The grey line represents the best fit with a logistic equation giving an IC50 and a slope of 3.4 ± 0.2 nM and 1.1 ± 0.05, respectively. (**B**) A representative hERG1 current in control (black line) and after application of CmERG1 toxin at 300 nM (light grey trace) and 3 nM (grey trace and red trace for resized current). (**C**) Time course of block during toxin application at 1 µM and during washout. Stimulus was a single depolarized step at 60 mv followed by a hyperpolarized step at −120 mV, applied every 5 s. (*n* = 3 for data recorded at 1 μM and 10 nM; *n* = 5 for data at 300 nM; *n* = 4 for data at 100 and 30 nM; *n* = 6 for data at 1 and 3 nM).

**Figure 4 toxins-13-00407-f004:**
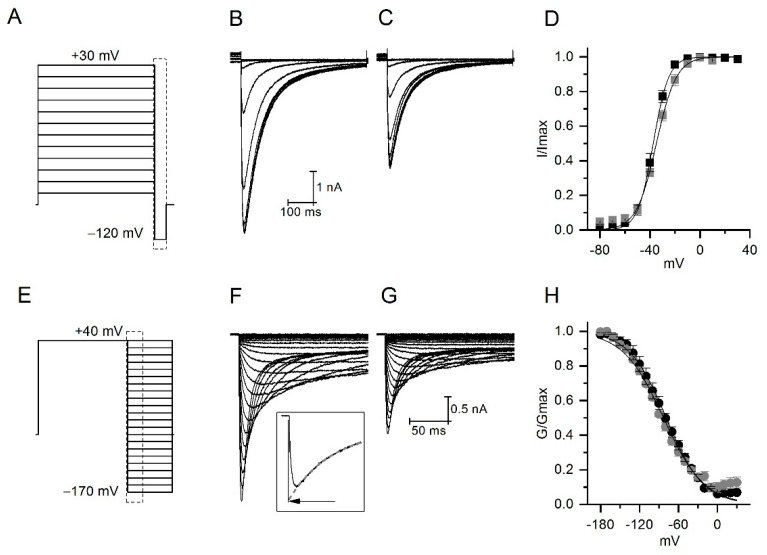
Effect of CmERG1 at 4 nM on the voltage dependence of activation and inactivation. (**A**) Stimulation protocol used to study the voltage dependence of activation. Tail currents were elicited by 5 s preconditioning steps ranging from +30 to −80 mV, followed by a 500 ms pulse at −120 mV. (**B**,**C**) typical example of tail currents recorded in control conditions and during application of CmERG1, respectively. (**D**) Voltage dependence of activation. Normalized peak currents are plotted against the voltage of the preconditioning pulses in control conditions (black squares) and in the presence of 4 nM CmERG1 (grey squares). Black lines are the best fit obtained to a Boltzmann equation giving V1/2 for activation of −37.5 ± 1.4 mV and −34.8 ± 0.8 mV and slope of 6.0 ± 0.3 and 8 ± 0.7 for the control and in the presence of 4 nM CmERG1, respectively (mean ± SE and *n* = 5, analyzed by paired sample *t*-test at 0.05 level). (**E**) Stimulation used to study the voltage dependence of inactivation. Currents were elicited by a 1 s preconditioning pulse at 40 mV followed by depolarization steps ranging from −170 mV to 40 mV. Currents recorded at potentials lower than −90 mV were corrected for deactivation as shown in the inset of panel (**F**). The deactivation process was fitted by a single exponential (grey dash line) extrapolated to the zero point of the depolarization step, indicated by the arrow. (**F**,**G**) typical example of currents recorded in control conditions and in presence of 4 nM CmERG1 respectively. (**H**) Voltage dependence of inactivation. Conductance was extrapolated from corrected peak currents, normalized, and then plotted versus voltage. Black lines are the best fit obtained by a Boltzmann equation giving V_1/2_ for inactivation of −77.6 ± 3.6 mV and −79 ± 1.3 mV and slope of 28.2 ± 2.2 and 29 ± 1.5 for control conditions and in presence of 4 nM CmERG1, respectively (mean ± SE and *n* = 5, analyzed by paired sample *t*-test at 0.05 level).

**Figure 5 toxins-13-00407-f005:**
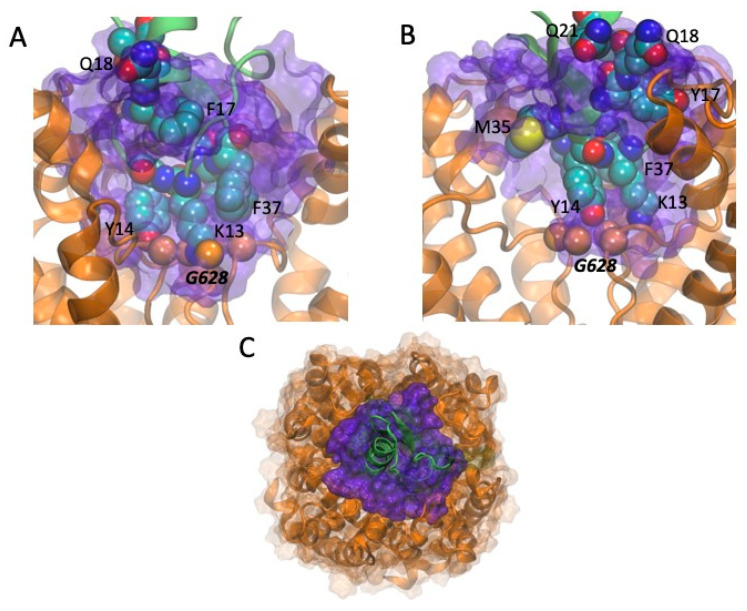
(**A**,**B**): Two possible binding modes that block the access to the selectivity filter. The channel pore is shown as an orange ribbon and the toxins as green ribbons. The contact volume or “seal” between the channel and the toxin (with a 4.5 Å cutoff) is shown as a translucent purple surface. Residues in the toxins that have been shown to affect binding or that are different between CnERG1 and CmERG1 (see Figure 2) are shown as spheres in CPK colors (C in cyan, O in red, N in blue, S in yellow). (**A**) In a high affinity pose, CmERG1 K13 penetrates the entry to the selectivity filter (marked by the orange spheres labeled as G628), flanked by Y14, F37, and F36 (behind K13 and F37), making a hydrophobic and cationic plug. (**B**) In a lower affinity pose for CnERG1, Y14 lies directly over the entry to the selectivity filter (marked by the orange spheres labeled as G628), while K13 engages in hydrogen bonds with two of the G628 carbonyls. (**C**) Top view of the CnERG1 complex, showing that one toxin is enough to fill the pore entrance of the channel.

**Figure 6 toxins-13-00407-f006:**
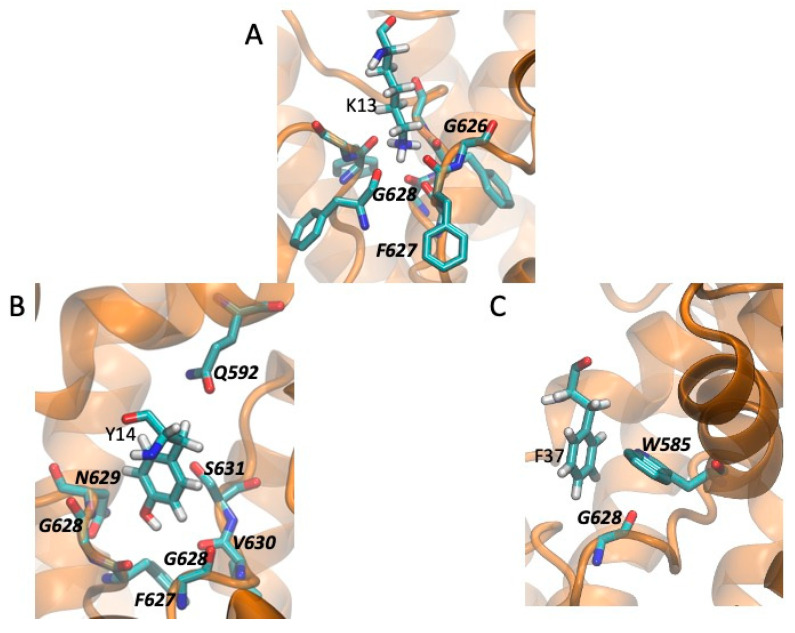
Channel-CmERG1 interactions in a high affinity pose for residues that impair binding when mutated (from Figure 2). The channel is displayed as an orange ribbon and the residues in licorice and CPK colors. Toxin residues include hydrogen atoms; channel residues are labeled in boldface and italics. Residues that interact with the selectivity filter; (**A**) K13 engages in hydrogen bonds with three of the four mainchain carbonyls of the F627 residues and van der Waals interactions with G626 and G628, blocking the pore. (**B**) Y14 is nested in a crevice on the side of the entrance to the selectivity filter, making van der Waals contacts with residues from two adjacent subunits. (**C**) F37 stacks in a T conformation against W585 and makes van der Waals contacts with the carbonyl of G628 at the entrance of the selectivity filter. Residues that interact with turret residues and contribute to the “seal” between the toxin and the channel: (**D**) F17 interacts with Y597 in a T conformation. (**E**) M35 is surrounded by the sidechains of S581, R582, and N588. (**F**) Q18 stacks against the peptide bond between Y597 and P596, and hydrogen bonds to R582 from the adjacent subunit. This same R also hydrogen bonds to the carbonyl of P596.

**Figure 7 toxins-13-00407-f007:**
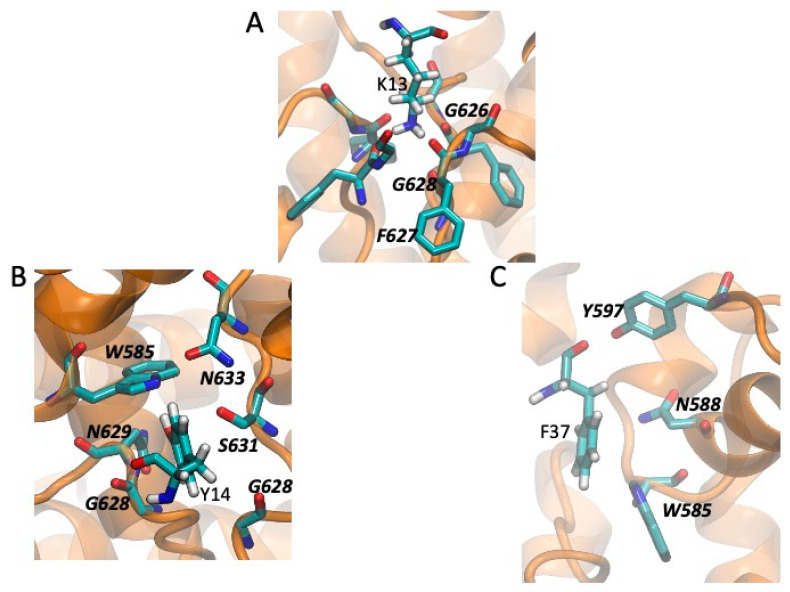
Channel-CnERG1 interactions in a high affinity pose for residues that impair binding when mutated (from Figure 2). The channel is displayed as an orange ribbon and the residues in licorice and CPK colors. Toxin residues include hydrogen atoms; channel residues are labeled in boldface and italics. Residues that interact with the selectivity filter: (**A**) K13 engages in hydrogen bonds with three of the four mainchain carbonyls of the F627 residues and van der Waals interactions with G626 and G628, blocking the pore. (**B**) Y14 is nested in a crevice on the side of the entrance to the selectivity filter, making van der Waals contacts with residues from two adjacent subunits and a hydrogen bond to W585 or S631. (**C**) F37 stacks in a parallel conformation against W585 and N588, while making a hydrogen bond with Y597 with its carbonyl. Residues that interact with turret residues and contribute to the “seal” between the toxin and the channel: (**D**) Y17 interacts with Y597 in a T conformation. (**E**): Q18 stacks against Y597 and the backbone of K595, and hydrogen bonds to H578 and R582 from the adjacent subunit. (**F**) Q21 is nested against the hydrogen bond between R582 and N598, engaging also in a T interaction with H578. (**G**) M35 is surrounded by the sidechains of S581, R582, Y597, and N588.

**Table 1 toxins-13-00407-t001:** Mass-spectrometry analysis of the most abundant peaks obtained by RP-FPLC.

Fraction	Molecular Mass (Da)
F13	**3704.97**
F14	**3718.19**
F15	**3543.74/4178.82/3228.7**
F16	**3571.8/4191.71**
F17	**3572.21**
F18	**4177.69**
F19	**4176.97**
F20	**2820.1/3980.01**
F21	**2819.8/3979.7**
F22	**4916.12**/6898.7/7942.6
* F23	6838.9/**4792.1**
F31	6793.37/**3376.05**

According to their molecular mass, the possible peptides active on potassium channels are indicated in bold. The fraction active on hERG1 channels is indicated with an asterisk. Molecular masses represent average masses of the peptides.

**Table 2 toxins-13-00407-t002:** ESI-MS analysis of the chromatographic fractions obtained by RP-HPLC from the sub-fractions from FII.6 to FII.10.

Fraction	Retention Time (min)	Molecular Mass (Da)
FII-6	24.5	2820.5
***FII-6***	***27.4***	***4792.8***
FII-6	28.7	4045.7
FII-6	29.1	4309.5; 3994
FII-6	32.3	6992
FII-6	33.4	4177; 6470
FII-6	34.6	6613.5; 6868.1
FII-6	35.4	6724
FII-7	20.66	3229
FII-7	28.4	4045.6
FII-7	29.2	6838.8
FII-8	23.2	4177.5
FII-8	25.2	3980.2
FII-8	29.2	8475.1; 4133.5
FII-8	37.9	7547.6; 7998.1
FII-9	21.6	3704.2; 3815.1
FII-9	26.1	4178.4
FII-9	26.4	8475.8; 4178
FII-9	27	7498.0; 7620.7
FII-9	27.2	7497.4; 7618
FII-10	18.7	3703.7; 3718.2
FII-10	23.2	8441; 5027.8; 4915
FII-10	28.6	7497; 7620

The fraction, retention time, and molecular mass of CmERG1 toxin are indicated in bold and italics. Molecular masses represent average masses of the peptides.

## Data Availability

The original electrophysiology data files presented in this study are available on request from the corresponding author.

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
