# Peer review of "Colombian Scorpion *Centruroides margaritatus*: Purification and Characterization of a Gamma Potassium Toxin with Full-Block Activity on the hERG1 Channel"

_toxins, 2021, doi:10.3390/toxins13060407_

Round 1

Reviewer 1 Report

The revised version looks ready for publication.

Author Response

Thank you for your positive response.

Reviewer 2 Report

The article has undergone a reorganization, which deserves to be acknowledged.

However, several points need to be clarified:

  • I requested on the first round to add a statistics section in materials and methods, and it has not been done. You have to indicate which statistical test has been used (L194 & L215).

  • The number of cells used in electrophysiology is only indicated in Figure 4H. You must indicate the n cells used in each assay (Figures 3 & 4) in both Mat & meth + captions under the Figures.

  • Shown in Figs 3 & 4 are the tail currents recorded during the hyperpolarizing step. Nothing is indicated nor discussed concerning the whole current recorded during the depolarizing step (I assume an outward current). This must be added and discussed in a Figure (or a supplementary Figure), even if the toxin does not affect the kinetics of the channel opening.

  • Please add a section “cells” in Materials and methods, to describe how CHO cells are cultured and how they are transfected to overexpress hERG1 channels. Indicate the protocol used for transfection (plasmid, lipofectamine etc.).

The article could be illustrated with an image of the scorpion Centruroides margaritatus.

Author Response

The article has undergone a reorganization, which deserves to be acknowledged.

However, several points need to be clarified:

  • I requested on the first round to add a statistics section in materials and methods, and it has not been done. You have to indicate which statistical test has been used (L194 & L215).

We apologize for not having attended your request exactly in the first round. We now divided the Electrophysiology subsection of the Materials and Methods into subsubsections: “5.6.1. Cells and solutions; 5.6.2. Patch-clamp recordings and data analysis” and “5.6.3. Statistical analysis”. As you suggested we added: Statistical analysis”. In this latter subsection, we moved the sentence: “Off-line analysis and graphs were performed by using Clampfit 10 (Molecular Devices) and Origin 8 (OriginLab, MA, USA). Where it is not otherwise indicated, electrophysiological data represent the mean of 3-6 cells ± standard error (S.E.)” and we added: “Each cell was recorded in absence and in presence of the toxin and the difference between these two conditions was analyzed by means of the paired sample t-test at 0.05 level.”

  • The number of cells used in electrophysiology is only indicated in Figure 4H. You must indicate the n cells used in each assay (Figures 3 & 4) in both Mat & meth + captions under the Figures.

In Figure 3 we added the sentence: (n=3 for data recorded at 1 µM and 10 nM; n=5 for data at 300 nM; n=4 for data at 100 and 30 nM; n=6 for data at 1 and 3 nM)”. In Figure 4, as you noticed, the number of cells are indicated for panel H but also this was already indicated for panel D (see L204: “mean ± SE and n=5”). We added at this point the sentence “analyzed by paired sample t-test at 0.05 level” that was missing in panel D. In Materials and methods, as we explained above, we added in the new subsubsection “5.6.3. Statistical analysis” the sentences: “Where it is not otherwise indicated, electrophysiological data represent the mean of 3-6 cells ± standard error (S.E.). Each cell was recorded in absence and in presence of the toxin and the difference between these two conditions was analyzed by means of the paired sample t-test at 0.05 level.”

  • Shown in Figs 3 & 4 are the tail currents recorded during the hyperpolarizing step. Nothing is indicated nor discussed concerning the whole current recorded during the depolarizing step (I assume an outward current). This must be added and discussed in a Figure (or a supplementary Figure), even if the toxin does not affect the kinetics of the channel opening.

 We prepared a supplementary figure (Figure S1) showing the whole current recorded and shown in Figure 4 (activation and inactivation protocols). In the subsubsection “Patch-clamp recordings and analysis” in “Material and Methods, Electrophysiology” section, we explain why we recorded the tail currents with the sentence: “During the depolarizing steps, hERG1 currents are usually small (Figure S1). This is a consequence of the very slow activation kinetics (in seconds) and the very fast inactivation kinetics (in milliseconds) of the channel that, for this reason, can be considered an inward rectifier. Therefore, it is a common practice to record the tail currents during the hyperpolarization, where the inactivation is quickly removed and the deactivation occurs slowly [48].

  • Please add a section “cells” in Materials and methods, to describe how CHO cells are cultured and how they are transfected to overexpress hERG1 channels. Indicate the protocol used for transfection (plasmid, lipofectamine etc.).

Following your recommendation, the description of the cell culture is now in the subsubsection: 5.6.1. Cells and solutions”. In this section, we added a brief description of how the CHO cell line, stably expressing hERG1 channels, was previously produced in our lab. CHO cells stably expressing hERG1 potassium channels were used for the electrophysiological experiments (hERG1 accession number: NP_000229). We previously prepared this cell line stably transfecting CHO cells with plasmid pcDNA3.1-hERG1 (a kind gift from Enzo Wanke from University of Milano-Bicocca, Italy). Briefly: CHO cells at 80% confluence in a 35 mm culture plate were transfected with 2 µg of pcDNA3.1-hERG1 mixed with 7uL of Lipofectamine (Invitrogen), accordingly to the manufacturer instructions. After 3 days, cells were selected by adding to the culture medium 2 mg/mL of G418 (SIGMA). After 10 days of selection, cells were cloned by limiting dilution and the resulting clones were probed for their current expression by electrophysiological recordings.” We also added the sentence: “A concentrated 100-1000x stock was prepared dissolving lyophilized toxin in distilled water and stored at -20°C until used (no more than 3 weeks).”

The article could be illustrated with an image of the scorpion Centruroides margaritatus.

Thank you for the suggestion. We included a picture of the scorpion Centruroides margaritatus in the graphical abstract.

Round 2

Reviewer 2 Report

Authors have brought modifications as requested.

Their paper is interesting but its main weakness stands in the very low number of cells they have been working with in electrophysiology (n=3, n=4...). This should be clearly highlighted as a burden for the whole study.

Author Response

Response to the review #2

Authors have brought modifications as requested.

Their paper is interesting but its main weakness stands in the very low number of cells they have been working with in electrophysiology (n=3, n=4...). This should be clearly highlighted as a burden for the whole study.

Thanks again for your constructive comments and suggestions. We added a sentence in the “Conclusions” section that clearly highlighted the low number of cells used in electrophysiology experiments: “This work have the weakness of a low number of cells we have been working with in electrophysiology (in some cases n = 3 or 4), nevertheless our results suggest that the CmERG1 toxin-channel interaction resembles the blocking....”

We hope you now consider this work suitable for publication

This manuscript is a resubmission of an earlier submission. The following is a list of the peer review reports and author responses from that submission.

Round 1

Reviewer 1 Report

Peer review of Toxins-1178403-v1

General comments

This manuscript describes some electrophysiological characterisation of venom from the scorpion Centuroides margaritartis, and one of its components, the peptide CmERG1. For the most part it is a good piece of work and has been carefully prepared, and contains some insights into the mechanisms of gamma-KTx which are quite interesting. I am not an expert in molecular modelling but I think that much of section 2.7 is quite speculative, however the models and data presented do lead to testable hypotheses, which is perhaps more important. Overall I suggest it is suitable for publication in Toxins after minor revisions to address the following specific comments:

Specific comments

  • Line 16: "member number" should be "member of number"
  • Lines 40 and 41 and perhaps elsewhere: KDa should be kDa
  • Line 110, 128, 133, 139, 152, 183, 376, 386, 468 and elsewhere: Italicise binomial name
  • Lines 99: please add a reference to illustrate the typical nature of these effects
  • Line 107: please insert a reference to another peptide with dual alpha and beta effects (there are several)
  • Line 135: should herg1 be hERG1?
  • Table 1: please indicate in the table legend or column names whether these masses represent peak masses, average masses, or monoisotopic masses, and if they represent peptide or ion masses
  • Lines 149–150: could you reword this to make it clear it is in reference to the previously proposed idea of KTx being 3-5 kDa?
  • Figure 3: does the red arrow in C and D indicate testing revealed this to be the functional fraction? If so, please state this
  • Lines 190–193: Please reword these two sentences which are very confusing:

"Primary structure determination of the CmERG1 was achieved by directly sequencing the native peptide, as well as by sequencing a reduced and alkylated sample of the same peptide, without the enzymatic digestion." After having read the methods I take it this is referring to the Edman sequencing (already mentioned on lines 175–177), rather than mass spec but it is very confusing, please reword.

"CmERG1 results in a 42 amino acids toxin with eight cysteines and four putative disulfide bonds" should perhaps be "CmERG1 is a toxin consisting of 42 amino acids with eight cysteines and four putative disulfide bonds" or similar

  • Line 206: "highlighted by yellow shadows" should be "highlighted in yellow"
  • Line 407: Do you have the exact monoisotopic mass for this peptide? If it exactly matches margatoxin that would be additional evidence.
  • Methods: Please indicate how the venom concentration was measured or estimated
  • Line 482: acetonitrile does not require capitalisation
  • Line 545–553 and perhaps elsewhere: the 2 should be subscripted in CO2, CaCl2, MgCl2, and others
  • References: Please check for consistency of capitalisation e.g. refs 9 and 10 are different

Reviewer 2 Report

The article titled “Colombian Scorpion Centruroides margaritatus: Venom Characterization and Identification of a Gamma Potassium Toxin with Full-Block Activity on the hERG1 Channel” looked through a multidisciplinary approach to the effects of this scorpion's venom. In electrophysiology (whole cell), the authors show that the venom acts on Nav channels by shifting the activation curve to more negative potentials, and by effectively inhibiting the amplitude of the current. On some Nav channels, the venom also induces an inhibition of inactivation. The venom is also active on certain K+ channel subtypes, notably Kv1.1 and ERG1. The toxin responsible for this inhibitory effect on ERG channels - CmERG1- is characterized by chromatography and mass spectrometry, and sequenced. It is a peptide of 42 amino acids and 4 2S bridges. The molecular interaction of the toxin with the channel is proposed using a docking model, to identify the residues involved in this interaction.

This study proposes the isolation and characterization of a toxin active on ERG channels, which makes it unique. The expression in the English language needs to be improved as many mistakes weaken the style. In addition, on the electrophysiological part, the authors must make important modifications before being able to publish this study.

Figures 1 and 2 : for better reading, please show the current trace + venom in red (and not grey), and same for the I/V curves.

Figure 1: you must indicate for each panel (A-G) how many cells were recorded. I did not find it in the draft. Please add this information in the caption of the Figure. I do not see the error bars for each point, nor any statistical analysis. You must include (i) error bars and (ii) a statistics section in materials and methods, and (iii) propose for each I/V curve a statistical comparison for each point between control and +venom. The n cells needs to be indicated for each Figure with electrophysiology (1-2; 5-6).

Why did you choose to express some Nav channels in HEK cells, and Nav1.7 + Kv and ERG channels in CHO cells? Don’t you think it might bias your analysis?

L105-107 and Figure 1: if the venom slows down the inactivation of Nav 1.4, 1.5 and 1.7, it must be quantified through either the I/V curve with steady-state inactivation, or through an histogram showing the t1/2inactivation. As such, Figure 1 is incomplete.

Figure 2: please add the error bars on 2F (Kv1.1; Kv1.7; hEAG1).

Minor

Centruroides margaritus should be written everywhere in italics. It is not always the case.

L39: “have been classified according to their targets into”. Scorpion toxins target not just Na+, K+ or RyR channels. Some toxins have been shown to block different Cav channels as well.

L64: “LD50 of 59.9 mg/kg“ Please specify, when proposing a LD50, the animal species and the route of administration (iv, ip, im…). Same apply for L380 and L381.

L96: sodium channel subtypes

L99: β toxins

L104 : α scorpion toxins

L116: scalers ?

L122: completely blocks

L200: Centruroides noxius

L227 : scaled current ?

L232 : concentration close to the IC50

L251-270:            A:            B and C                D:           E)            F and G)               H)

Please homogenize.

L276: shown by both toxins

Round 2

Reviewer 2 Report

Although this article is interesting and contains quite original results, the fact that it was carried out, for the first two figures, with an insufficient number of cells, does not allow it to be published as is. These data, realized on n=1 or n=2 cells are only preliminary results.

I can only recommend to the authors to complete the number of cells on which they have generated these data to have a good repeatability, attested by an adequate statistical test. Alternatively, they should remove the first two figures from the paper and restructure it accordingly.

One of the major pitfalls of in vitro or in vivo experimentation is precisely the poor repeatability of a protocol. In this case, this paper cannot be published in its present form.